# SeeingEye: Agentic Information Flow Unlocks Multimodal Reasoning in Text-Only LLMs

## Abstract

Recent advances in text-only large language models (LLMs), such as DeepSeek-R1, demonstrate remarkable reasoning ability. However, these models remain fragile—or entirely incapable—when extended to multimodal tasks. Existing approaches largely rely on single-form captions, which lack diversity and often fail to adapt across different types of Visual Question Answering (VQA) benchmarks. As a result, they provide no principled or efficient channel for transmitting fine-grained visual information. We introduce **SeeingEye**, a modular framework that unlocks multimodal reasoning in text-only LLMs through an **agent-based small VLM translator**. This translator acts as a perception agent: it can invoke specialized tools (*e.g.*, OCR and crop) and iteratively distill multimodal inputs into structured intermediate representations (SIRs) tailored to the question. These SIRs are then passed to the text-only LLM, which serves as a reasoning agent. Crucially, the translator and reasoner engage in multi-round feedback and interaction, enabling the extraction of targeted visual details and yielding more confident answers. Experiments on knowledge-intensive VQA benchmarks, including MMMU and MIA-Bench, demonstrate that Translation First not only reduces inference cost but also surpasses much larger end-to-end VLMs. For example, an instantiation combining a 3B-parameter vision translator with an 8B-parameter language reasoner outperforms a monolithic 32B VLM on challenging knowledge-based questions. Our results highlight that decoupling perception from reasoning via agent information flow offers a scalable and plug-and-play pathway to multimodal reasoning, allowing strong text-only LLMs to fully leverage their reasoning capabilities.

## 1 Introduction

Recent text-only LLM reasoners, such as DeepSeek-R1, have demonstrated remarkable text-only reasoning, pushing the frontiers of artificial intelligence in tasks from code generation to complex problem-solving Brown et al. (2020); Guo et al. (2025); Liu et al. (2025); Han et al. (2025). Compared to multimodal reasoners, they enjoy a wide adoption and cost efficiency, but lack multimodal reasoning capabilities. It thus exists a central research question: can we bridge text-only LLM reasoners with multimodal reasoning capabilities that are effective and more cost-efficient than multimodal reasoning models?

A common paradigm to answer this question has centered on converting the visual input into text. Early approaches relied on generating static, single-form captions, from general descriptions to more query-focused variants Khademi et al. (2023); Özdemir & Akagündüz (2024); Ma et al. (2024). However, these non-interactive descriptions lack the adaptability for diverse VQA tasks and create a fixed information bottleneck. Recognizing this, more recent works introduce dynamic primitives like tool use Wu et al. (2023) or integrated active perception Wu & Xie (2024). While a significant step forward, these methods present new limitations: the information flow is often an unstructured conversational history of tool calls, or the perception and reasoning modules are tightly coupled within monolithic VLMs. Such architectures are difficult to scale and cannot easily leverage the distinct, rapidly advancing power of state-of-the-art text-only reasoners. Consequently, even advanced

systems still lack a formal, structured medium for information exchange—an efficient channel that allows a powerful, text-only reasoning agent to iteratively query and comprehend visual information.

Motivated by these limitations, we argue the key to unlocking multimodal reasoning in text-only LLMs is not to simply describe, but to actively *translate*. We introduce **SeeingEye**, a novel, modular framework that reconceptualizes the vision component as an agent-based translator rather than a passive descriptor. The Translator interacts with the visual input by invoking specialized tools, such as OCR for text extraction or cropping for targeted inspection, to iteratively distill the complex scene into a novel Structured Intermediate Representation (SIR) (See Fig. 3) to preserve as much valuable information as possible across modalities. Then, based on the input question, the Translator automatically selects appropriate tools and dynamically adjusts its execution steps, ultimately generating the SIR in various forms tailored to the problem-solving process. Crucially, the process is not unidirectional; the reasoning agent can provide feedback to the Translator, requesting clarifications that prompt further tool use to refine the SIR. This multi-round interaction creates a targeted information flow that extracts the precise visual evidence needed to arrive at a confident answer.

Through comprehensive experiments on knowledge-intensive VQA benchmarks like MMMU and MIA-Bench, we demonstrate that our agent-based, modular system (*e.g.*, a 3B VLM translator + 8B LLM reasoner) not only reduces inference costs but also surpasses the performance of much larger, monolithic end-to-end VLMs (*e.g.*, a 32B model).

Our core contributions are as follows:

- We propose **SeeingEye**, a novel, **plug-and-play framework** that unlocks the multimodal reasoning capabilities of powerful, pre-existing text-only LLMs without requiring any modification to their architecture.

- **Structured Intermediate Representation** (SIR), creating a targeted information channel that delivers precise visual evidence to the text-only reasoner.

- We design a novel **Agentic Information Flow**, where a translator agent autonomously selects tools based on the VQA task. This multi-round interaction generates and refines the SIR, being effective and cost-efficient.

Overall, our results highlight a scalable pathway to advanced multimodal reasoning, liberating strong text-only LLMs to fully leverage their powerful reasoning capabilities on visual data. We are releasing our code to facilitate future research in this direction.

## 2 RELATED WORK

**The Evolving Landscape of Visual Question Answering.** The VQA landscape has rapidly evolved from simple recognition towards multifaceted reasoning. While benchmarks like GQA Hudson & Manning (2019) introduced critical challenges in compositional and spatial reasoning, a recent wave of datasets tests more specialized, expert-level capabilities. These include reasoning with college-level knowledge (MMMU Yue et al. (2024a)), across multilingual contexts (M3Exam Zhang et al. (2023b)), through complex, layered instructions (MIA-Bench Qian et al. (2024)), and within specific domains like chart analysis (EncQA Mukherjee et al. (2025)). This task diversity exposes the limitations of monolithic models that use a fixed visual encoding strategy Agrawal et al. (2022); Ke et al. (2025). Such static approaches create an information bottleneck, failing to distill the precise visual details required for each unique challenge and thus motivating our adaptive, agent-based translation framework.

**Structured Representations for Multimodal Reasoning.** A critical bottleneck in existing multimodal systems is the conversion of rich visual scenes into coarse textual summaries, such as generic captions or unordered OCR text Alayrac et al. (2022); Li et al. (2023), which discards vital semantic and spatial relationships. Insights from the text-only domain have shown that structured inputs—such as JSON schemas, key-value pairs, or knowledge graphs Cheng et al. (2024); Sun et al. (2023)—significantly enhance an LLM's reasoning capabilities by making relationships explicit and reducing ambiguity. Our work operationalizes this principle for the visual domain. We propose the Structured Intermediate Representation (SIR) as a rich, deliberate communication channel that bridges the gap between visual perception and high-fidelity reasoning.

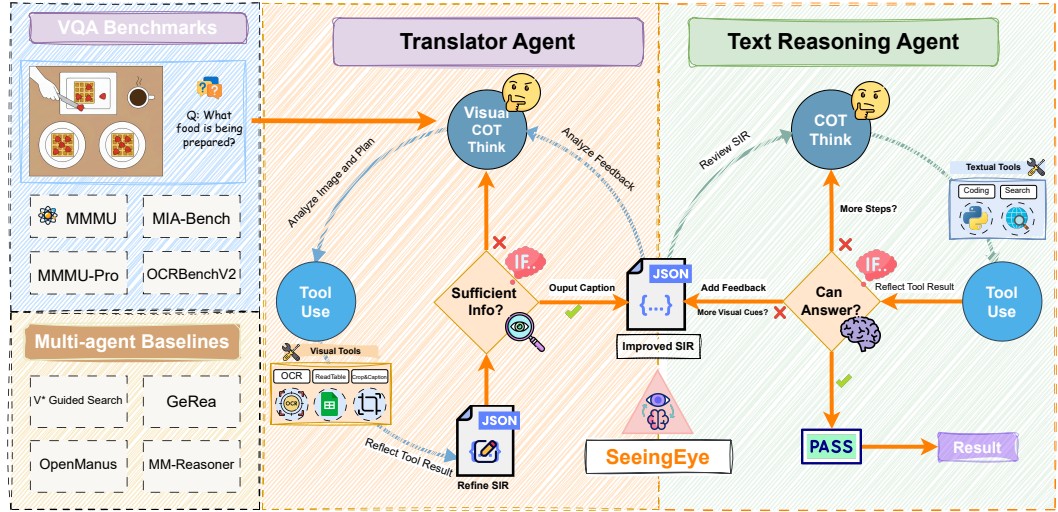

Figure 1: The Agentic Information Flow of our Translation First framework. The process begins with the **Translator Agent** (left), which takes the previous state's SIR and external feedback to perform a Visual Chain-of-Thought (VCoT) analysis. It uses tools to gather new visual evidence, reflects on the results, and iteratively refines the SIR. If the information is deemed sufficient (**PASS**), the improved SIR is passed to the **Text Reasoning Agent** (right). The Reasoner performs its own CoT-driven analysis and tool use. Based on its confidence, it either produces a final answer (**PASS**) or generates targeted feedback (**FAIL**), initiating a new outer loop iteration for the Translator to gather more specific visual cues.

**Agentic Frameworks and Visual Chain-of-Thought.** LLM-based agents have demonstrated powerful abilities in planning, tool use, and interactive reasoning Yao et al. (2023); Shinn et al. (2023). This agentic paradigm, often augmented by Chain-of-Thought (CoT) prompting Wei et al. (2022) to improve reasoning transparency, has been extended to the multimodal domain. In Visual CoT, models generate step-by-step textual rationales to ground their reasoning in visual evidence Zhang & Zhang (2023); Rose et al. (2023). Our work advances this concept from generating linear, unstructured rationales to a more sophisticated, agent-driven process. The translator agent in our *Translation First* framework engages in a multi-round *Agentic Information Flow*, dynamically constructing and refining a *structured* representation (our SIR) through a feedback loop with the reasoner, enabling a more targeted and adaptive problem-solving strategy.

## 3 METHOD

Our proposed framework, **SeeingEye**, unlocks the multimodal reasoning capabilities of text-only LLMs by introducing a decoupled, two-agent system. This system comprises a **Translator Agent** ($\mathcal{A}_T$) and a **Reasoning Agent** ($\mathcal{A}_R$). The core of our method lies in a novel **Agentic Information Flow**, orchestrated through a nested loop structure, where the agents collaborate by iteratively improving a central **Structured Intermediate Representation (SIR)**. Figure 1 provides a high-level illustration, while Algorithm 1 presents the formal specification of this interactive process.

### 3.1 THE TRANSLATOR AGENT: GROUNDED VISUAL ANALYSIS

The Translator Agent, $\mathcal{A}_T$, is a lightweight VLM responsible for converting raw pixel data into a rich, structured, and query-relevant format. Its state at the start of outer-loop iteration $i$ is defined by the input image $I$, the question $Q$, the SIR from the previous iteration $S_{i-1}$, and feedback from the Reasoner $F_{i-1}$. The agent's goal is to produce an improved SIR, $S_i$, through a multi-step inner loop.

---

**Algorithm 1** SeeingEye: Agentic Information Flow

---

1: **Input:** question $Q$, options $O$, image $I$
2: **Parameters:** MAX_ITERS, MAX_STEPS
3: **Initialize:** $S_0 \leftarrow$ null, $A_{\text{final}} \leftarrow$ null
4: **for** $i = 1 \rightarrow$ MAX_ITERS **do**
5:                                                ▷ *— Translator Agent Inner Loop —*
6:     $S_{\text{current}} \leftarrow S_{i-1}$
7:     **for** $j = 1 \rightarrow$ MAX_STEPS **do**
8:         $a_T \leftarrow$ TranslatorPolicy(VCoT($I, Q, S_{\text{current}}$))
9:         **if** $a_T$ is **ToolCall**($o_T$, args) **then**
10:             $r_T \leftarrow$ ExecuteTool($o_T$, args)
11:             $S_{\text{current}} \leftarrow$ RefineSIR($S_{\text{current}}, r_T$)
12:         **else if** $a_T$ is **TerminateSIR then**
13:             **break**
14:     $S_i \leftarrow S_{\text{current}}$
15:                                              ▷ *— Reasoning Agent Inner Loop —*
16:     **for** $k = 1 \rightarrow$ MAX_STEPS **do**
17:         $a_R \leftarrow$ ReasonerPolicy(CoT($S_i, Q, O$))
18:         **if** $a_R$ is **ToolCall**($o_R$, args) **then**
19:             $r_R \leftarrow$ ExecuteTool($o_R$, args)
20:         **else if** $a_R$ is **TerminateAnswer**($A$) **then**
21:             $A_{\text{final}} \leftarrow A$; **goto** 27
22:         **else if** $a_R$ is **TerminateFeedback**($F$) **then**
23:             $S_i \leftarrow S_i \oplus F$
24:             **break**
25: **if** $A_{\text{final}}$ is null **then**
26:     $A_{\text{final}} \leftarrow$ ForceAnswer($S_{\text{MAX\_ITERS}}, Q, O$)
27: **return** $A_{\text{final}}$

---

**Visual Chain-of-Thought (VCoT) Analysis.** At each inner step $j$, the agent generates a *Visual Chain-of-Thought* (VCoT) Zhang et al. (2023a), a textual thought process $c_T^{(j)}$ describing its direct visual observations and its reasoning for the next action.

$$c_T^{(j)} = \text{VCoT}(I, Q, S_{i-1}, F_{i-1}, h_T^{(j-1)}) \tag{1}$$

where $h_T^{(j-1)}$ is the history of actions within the current inner loop.

**Adaptive Tool Selection and Execution.** Guided by its VCoT, $\mathcal{A}_T$ selects a tool $o_T^{(j)}$ from its toolset $\mathcal{O}_T$ via its policy $\pi_T$, and executes it to yield a result $r_T^{(j)}$.

$$o_T^{(j)} \sim \pi_T(c_T^{(j)}, I, Q, S_{i-1}, F_{i-1}, h_T^{(j-1)}) \tag{2}$$

Among the tools in $\mathcal{O}_T$, **SmartGridCaption** is a specialized sub-routine designed for complex spatial queries that require targeted analysis. As illustrated in our case study (Figure 2), the tool is invoked when a direct visual observation proves insufficient. Initially, the agent generates a global SIR describing a "church building" but cannot identify the "animal in the poster" from this coarse view (Step 1).

To resolve this ambiguity, the tool first discretizes the image into a $4 \times 4$ grid. It then leverages a vision-LLM to interpret the query and select the most informative patches, in this instance, the rectangular region [9, 9] containing the poster (Step 2). A detailed caption is then generated for this specific crop, yielding the critical observation: "Poster featuring a person holding a dove" (Step 3). Crucially, this new, fine-grained detail is not treated in isolation. As shown in the "Integration Process", it is strategically integrated with the previous global description to create an updated, more comprehensive SIR. The SIR thus evolves from a generic overview to a targeted representation containing the precise fact needed for the query. This procedure of targeted refinement effectively transforms a vague spatial query into a high-confidence textual fact, enabling the Reasoning Agent to deduce the final answer with ease (Step 4).

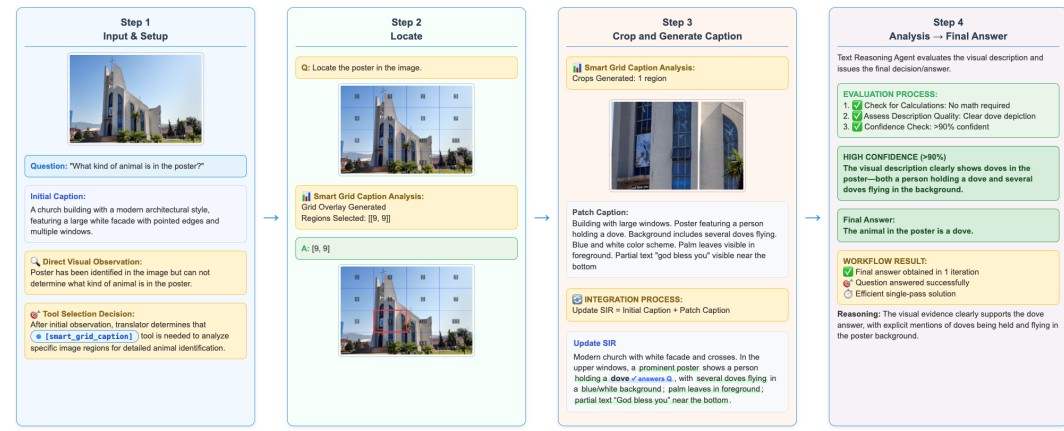

Figure 2: A detailed case study of the **SmartGridCaption** tool. (1) An initial global SIR fails to identify the animal in the poster. (2) The tool grids the image and locates the relevant patch [9, 9]. (3) A fine-grained patch caption is generated and integrated, updating the SIR with the crucial detail of a "dove". (4) This refined SIR enables the Reasoning Agent to provide a high-confidence final answer in a single iteration.

**Iterative SIR Refinement and Termination.** A core feature of our framework is the iterative refinement of the SIR within the Translator's inner loop. After each tool use, the agent reflects on its VCoT and the tool result to update the SIR. Let $S^{(j-1)}$ be the SIR at the beginning of the step; the refinement process is:

$$S^{(j)} = \text{RefineSIR}(S^{(j-1)}, c_T^{(j)}, r_T^{(j)}) \tag{3}$$

Following this refinement, the agent quantitatively assesses the completeness of the updated SIR, producing a confidence score $c_s^{(j)} = \text{AssessSufficiency}(S^{(j)})$. This score is compared against a predetermined sufficiency threshold, $\tau_T$. The inner loop terminates by invoking the `TerminateAndOutputSIR` tool if this confidence exceeds the threshold ($c_s^{(j)} \geq \tau_T$) or if the maximum step limit $N_T$ is reached. Upon termination, the agent outputs the final SIR for the outer loop, $S_i = S^{(j)}$, along with a categorical confidence level $C_T \in \{\text{low, mid, high}\}$ derived from the final score. If confidence is below the threshold and steps remain, the agent continues its inner loop to gather more visual information.

## 3.2 THE REASONING AGENT: HIGH-LEVEL COGNITION AND DECISION-MAKING

The Reasoning Agent, $\mathcal{A}_R$, is a powerful text-only LLM whose state for iteration $i$ consists of the SIR $S_i$, the question $Q$, and a short-term memory $M_{R,i-1}$ summarizing its prior actions. It leverages the SIR for high-level reasoning and to decide on a terminal action.

**SIR-Grounded Analysis and Tool Use.** The Reasoner initiates its own inner loop, generating a chain-of-thought $c_R^{(k)}$ and using its policy $\pi_R$ to select a tool $o_R^{(k)}$ from its distinct toolset $\mathcal{O}_R$ (e.g., CodeInterpreter, Search).

$$c_R^{(k)} = \text{CoT}(S_i, Q, M_{R,i-1}, h_R^{(k-1)}) \tag{4}$$

$$o_R^{(k)} \sim \pi_R(c_R^{(k)}, S_i, Q, M_{R,i-1}, h_R^{(k-1)}) \tag{5}$$

The tool result $r_R^{(k)}$ is appended to its inner-loop history $h_R^{(k)}$.

**Terminal Decision Policy.** The Reasoning Agent's decision-making is also governed by a confidence-threshold mechanism. After each inner-loop step $k$ (which includes its own CoT and optional tool use), the agent assesses its ability to answer the question based on its current reasoning history $h_R^{(k)}$, yielding a confidence score $c_a^{(k)}$. This score is compared against a high-confidence answering threshold, $\tau_R$.

- **If confidence is high** ($c_a^{(k)} \geq \tau_R$) or if the final outer iteration ($i = \text{MAX\_ITERS}$) is reached, the agent is compelled to execute the `TerminateAndAnswer` action. It generates the final answer $A$, and the process terminates.

- **If confidence is low** ($c_a^{(k)} < \tau_R$), the agent's decision policy $\pi_D$ makes an autonomous choice. It can either continue its inner reasoning loop (if $k < N_R$) to further analyze the SIR or use more textual tools, or it can execute the `TerminateAndAskTranslator` action. This choice is formalized as:

$$a_{\text{final}} \sim \pi_D(S_i, Q, h_R^{(k)}) \tag{6}$$

where $a_{\text{final}} \in \{\text{ContinueReasoning}, \text{TerminateAndAskTranslator}\}$. Choosing the latter synthesizes a feedback query $F_i$ specifying the missing visual information, which is passed back to the Translator Agent to initiate a new outer loop iteration.

## 4 EXPERIMENTS

We conduct a series of experiments to evaluate the effectiveness of our SeeingEyeframework. Our evaluation is designed to answer a central research question: How does our proposed Translator-based Agentic Information Flow, designed to unlock the reasoning capabilities of text-only LLMs, compare in terms of performance and efficiency against state-of-the-art monolithic VLMs and other advanced agent-based approaches?

### 4.1 EXPERIMENTAL SETTINGS

**Benchmarks.** We evaluate our framework on a suite of challenging, reasoning-centric Visual Question Answering (VQA) benchmarks that require deep understanding of visual details, text, and domain-specific knowledge.

- **MMMU** Yue et al. (2024a): A massive, multi-discipline multimodal benchmark featuring questions from college-level exams across six core disciplines, requiring expert-level knowledge and reasoning. We report on the validation set.

- **MMMU-Pro** Yue et al. (2024b): A more challenging successor to MMMU, curated by human experts to feature more complex reasoning chains and reduce annotation artifacts. We evaluate on both the Standard and Vision subsets.

- **OCR-BenchV2** Fu et al. (2024): A comprehensive benchmark for evaluating OCR capabilities in the wild, testing the model's ability to read and interpret text from diverse and complex scenes.

- **MIA-Bench** Qian et al. (2024): A Multimodal Instruction-following and Analysis benchmark designed to assess a model's ability to follow complex instructions that require comparing, calculating, and reasoning over multiple image regions.

**Baselines.** We compare our method against two categories of leading models:

- **End-to-End VLMs**: These are monolithic, state-of-the-art models that process image and text inputs in a unified architecture. We include several variants of **Qwen2.5-VL** Bai et al. (2023), a powerful series of VLMs, and **GPT-4o-mini**, a highly capable multimodal model from OpenAI Achiam et al. (2023). These models represent the dominant paradigm and serve as a direct point of comparison for overall performance.

- **Advanced Modular Frameworks**: We also compare against recent methods that share our motivation of moving beyond static image descriptions. **V\*** Wu & Xie (2024) incorporates a guided visual search mechanism to actively seek information within an image. **Open-Manus** mannaandpoem et al. (2025) is a comprehensive open-source toolkit for building multimodal agents. To create a robust baseline that isolates the impact of our agentic flow, we adapt this toolkit using **Qwen3-8B** as the base model. We integrate our own meticulously designed suite of prompts, textual tools, and termination logic, while equipping it with a powerful set of visual tools, even including the **Qwen2.5-VL** model as a callable visual analysis tool. These baselines allow us to compare different approaches to dynamic visual interaction and reasoning.

**Implementation Details** Our model, referred to as **SeeingEye**, is instantiated using a 3B parameter Vision Language Model (Qwen2.5-VL) as the Translator Agent ($\mathcal{A}_T$) and an 8B parameter text-only Large Language Model (Qwen3) as the Text Reasoning Agent ($\mathcal{A}_R$). For our experiments, the inner loops for both agents ($N_T, N_R$) and the outer feedback loop ($N_{\text{outer}}$) are each capped at a maximum of 3 iterations to ensure computational tractability.

## 4.2 Main Results

The main results of our experiments are presented in Table 1. Our SeeingEyeframework, despite utilizing a significantly smaller model combination (3B VLM + 8B LLM), demonstrates a commanding performance across a suite of challenging benchmarks.

Table 1: Performance comparison on various knowledge-intensive multimodal benchmarks. Our method, SeeingEye, utilizes a significantly smaller model size compared to the baselines. Scores are reported in accuracy (%).

| Method | MMMU$_{\text{val}}$ | MMMU-Pro$_{\text{std.}}$ | MMMU-Pro$_{\text{vis.}}$ | OCR-BenchV2 | MIA-Bench |
|---|---|---|---|---|---|
| Qwen 2.5-VL-3b | 48.33 | 25.82 | 24.61 | 33.33 | 76.9 |
| Qwen 2.5-VL-7b | 51.11 | 23.60 | 23.26 | 32.96 | 79.9 |
| Qwen 2.5-VL-32b | 51.56 | 32.93 | 28.77 | 33.49 | 89.6 |
| GPT-4o-mini | 55.00 | 38.99 | 31.72 | 33.17 | **91.4** |
| V* Guided Search | 14.78 | 11.79 | 10.40 | 13.50 | 66.3 |
| OpenManus | 50.67 | 18.18 | 16.15 | 33.65 | 82.4 |
| **SeeingEye(Ours)** | **60.78** | **44.62** | **33.33** | **33.99** | 84.1 |

**Superiority over End-to-End VLMs.** Our results reveal a clear trend: on complex, knowledge-intensive reasoning benchmarks like MMMU and MMMU-Pro, our modular framework consistently and significantly outperforms even the largest monolithic VLMs. This outcome challenges the prevailing paradigm that performance gains in multimodality are primarily driven by scaling up end-to-end architectures. We posit that the reasoning capabilities of powerful text-only LLMs are a distinct and highly valuable asset that is not fully leveraged in monolithic designs. By decoupling perception from reasoning, our framework allows the text-only agent to operate in its native domain, processing rich, structured textual information rather than latent visual features. This architectural choice proves to be a more parameter-efficient and effective pathway to unlocking advanced multimodal reasoning.

**Effectiveness of the Agentic Information Flow.** A more telling comparison is against advanced modular frameworks. Our robust OpenManus baseline—equipped with a powerful 8B text model and a full VLM as a callable tool—still falls remarkably short of our performance on reasoning-heavy tasks. This stark difference underscores a critical insight: for agent-based systems, the quality of the tools is secondary to the quality of the communication protocol between agents. Traditional agent frameworks often rely on passing unstructured, monolithic strings (*e.g.*, a long caption) or conversational history as the medium for information transfer. This approach treats the output of a visual tool as a final, static artifact. The Reasoning Agent cannot query specific attributes of this information, nor can it request targeted updates. In contrast, our **Agentic Information Flow** is mediated by the **Structured Intermediate Representation (SIR)**. The SIR is not merely a string; it is a stateful, mutable, and query-able data object. The Translator populates it with typed visual evidence, and the Reasoner can issue precise feedback to refine specific fields within it. This transforms the interaction from a simple, stateless exchange of text blobs into a high-fidelity, collaborative process, creating a communication channel that is fundamentally more effective for complex, multi-step reasoning.

## 5 DISCUSSION

### 5.1 PLUG-AND-PLAY REASONING AGENTS

To validate the plug-and-play nature of our framework, we conduct an ablation study by fixing the Translator Agent while varying the text-only Reasoning Agent ($\mathcal{A}_R$). As shown in Table 2, our framework is model-agnostic, successfully integrating with various open-source and proprietary reasoners.

Crucially, the overall system performance scales directly with the reasoning capability of the text-only model, improving from 52.67% with Qwen3-8B to 54.67% with the larger Qwen3-14B on the MMMU$_{dev}$ set. This result strongly validates our core hypothesis: the SeeingEyearchitecture effectively isolates and leverages the reasoning power of the text-only agent, confirming that our performance gains are fundamentally driven by the strength of the chosen reasoner.

Table 2: Performance on the MMMU$_{dev}$ set when varying the text-only Reasoning Agent. The SeeingEyeis kept fixed. Results show that system performance scales with the reasoner's capability.

| Benchmark | Qwen3-8B | Qwen3-14B | GPT-4o-mini (text-only) |
|---|---|---|---|
| **MMMU$_{dev}$** (%) | 52.67 | 54.67 | 54.29 |

### 5.2 THE IMPACT OF THE AGENTIC INFORMATION FLOW

To further isolate the contribution of our proposed Agentic Information Flow, we conducted a critical ablation study. We constructed a powerful baseline by embedding a top-tier monolithic VLM, **GPT-4o-mini**, within the **OpenManus** agentic framework. This baseline was equipped with our full suite of meticulously designed textual tools (*e.g.*, Python execution) and termination logic, effectively creating a "best-of-both-worlds" traditional agent. The goal was to test whether a powerful, tool-augmented VLM could match the performance of our decoupled system.

The results, shown in Table 3, are unequivocal. On the MMMU dev set, our method, which uses a much smaller Qwen3-8B text reasoner, outperforms the GPT-4o-mini-powered OpenManus agent by a significant margin (52.67% vs. 46.77%). This finding is profound. Even when a state-of-the-art VLM is given the agency to use tools, its monolithic nature imposes a fundamental limitation. The internal reasoning process of the VLM is a black box. It cannot externalize its visual understanding into a structured, query-able format that can be inspected or refined. The agent's decisions are based on latent representations rather than a transparent, symbolic state like our SIR.

This experiment validates our central thesis: the architectural separation of a visual *Translator* and a textual *Reasoner*, communicating via a structured and mutable SIR, is fundamentally more effective than simply augmenting a monolithic VLM with tools. The SIR acts as the critical bridge that allows for transparent, high-fidelity information exchange and collaborative refinement, a mechanism that is absent in even the most powerful tool-augmented VLM agents.

Table 3: Performance on the MMMU$_{dev}$ set, comparing our Translation First framework against a powerful baseline combining the OpenManus agentic framework with the GPT-4o-mini VLM.

| Method | MMMU Dev Accuracy (%) |
|---|---|
| OpenManus (GPT-4o-mini) | 46.77 |
| SeeingEye(Ours) | **52.67** |

### 5.3 THE EFFICACY OF MULTI-ROUND INTERACTION

A cornerstone of our Translation First framework is the multi-round interaction that allows the Reasoning Agent to provide feedback and the Translator Agent to iteratively refine the SIR. To quantify the impact of this mechanism, we conducted an ablation study on the maximum number of outer loop iterations on the challenging MMMU-Pro (Vision) benchmark.

As demonstrated in Table 4, the benefits of our iterative refinement process are substantial and unequivocal. In the single-iteration setting (Max Iterations = 1), where the Reasoner cannot provide feedback, the system achieves a baseline accuracy of 34.21%. This is analogous to static, one-shot captioning methods. Enabling a single round of feedback (Max Iterations = 2), where the Translator can act upon the Reasoner's request for more specific visual information, yields a significant performance gain to 36.84%.

Most notably, allowing for up to three full iterations elevates the performance to **44.62%**, an absolute improvement of over 10% compared to the single-shot approach. This steep performance curve provides strong empirical evidence for our central hypothesis: complex multimodal reasoning is not a monolithic perception task, but an iterative process of inquiry and refinement. The ability for the agents to repeatedly pass and modify the SIR allows the system to progressively drill down into the most critical visual details, discard initial ambiguities, and ultimately converge on a high-fidelity representation of the scene that is precisely tailored to the reasoning needs of the query.

Table 4: Performance on MMMU-Pro (Vision) when varying the maximum number of outer loop iterations. The results clearly demonstrate the significant benefit of multi-round SIR refinement.

| Benchmark | Max Outer Iterations | | |
|---|---|---|---|
| | 1 | 2 | 3 |
| MMMU-Pro (Vision) (%) | 34.21 | 36.84 | **44.62** |

## 6 CONCLUSION

In this work, we addressed the challenge of unlocking multimodal reasoning in powerful, pre-existing text-only LLMs. We introduced **SeeingEye**, a novel framework that decouples perception from reasoning through a collaborative, two-agent system. Our core innovation is the **Agentic Information Flow**, where a lightweight Translator Agent iteratively generates and refines a **Structured Intermediate Representation (SIR)** to provide targeted, high-fidelity visual evidence to a text-only Reasoning Agent. Comprehensive experiments demonstrate that our modular, plug-and-play approach is not only more cost-efficient but also significantly outperforms larger, state-of-the-art monolithic VLMs on complex reasoning benchmarks. Our findings suggest that the future of advanced multimodal AI may lie not in ever-larger end-to-end models, but in the structured, synergistic collaboration between specialized agents.

### ETHICAL STATEMENT

Our work focuses on developing a more efficient and transparent framework for multimodal reasoning, primarily evaluated on established, public academic benchmarks. We acknowledge that, like all powerful generative models, this technology could be misused for generating misleading or harmful content. However, our design inherently mitigates some of these risks. The Translator Agent is explicitly prompted to be a neutral, objective visual describer, reducing the likelihood of generating biased or speculative interpretations. Furthermore, the framework's application to knowledge-intensive, fact-based Visual Question Answering steers its use toward verifiable and grounded reasoning rather than open-ended, unconstrained generation. We used only publicly available models and datasets, ensuring no privacy violations. We believe the primary impact of our research will be positive, contributing to the development of more modular, interpretable, and scalable AI systems.

### LLM USE STATEMENT

LLM were used in two ways during this work. First, LLMs served as the backbone agents evaluated in our experiments (e.g., GPT-4o-mini, Qwen3-8B). Second, we used LLMs to assist with writing tasks such as grammar error checking, and formatting, but all conceptual contributions, experimental design, and analyses were performed by the authors.

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

## A APPENDIX

### A.1 CASE STUDY OF SIR

### A.2 PROMPT

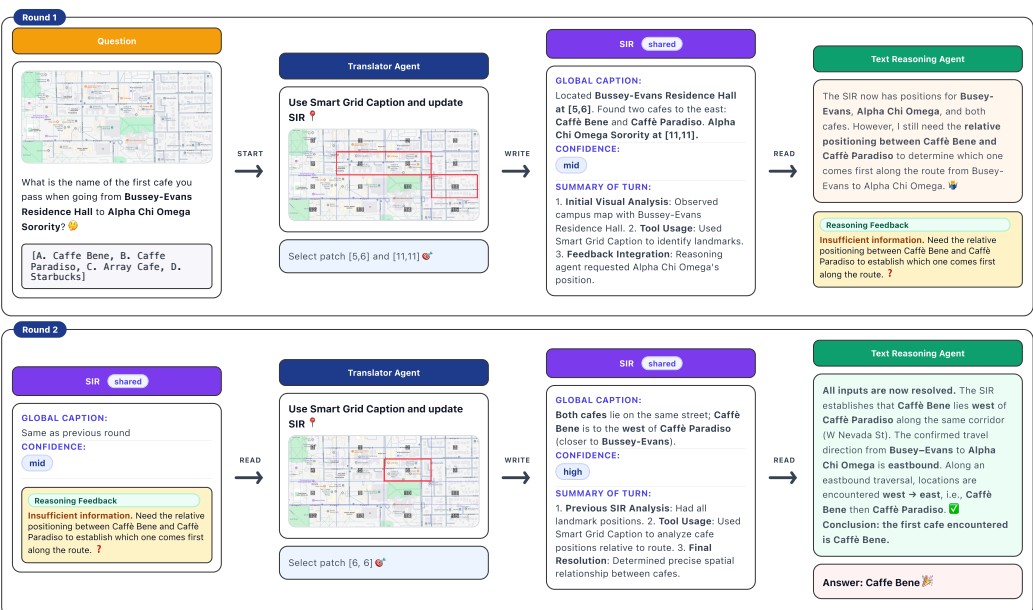

Figure 3: A Case Study of SIR.

---

**Translator Agent: System Prompt**

**Prompt Overview:** Guides the lightweight VLM to act as a Visual-Only Captioner. Its sole objective is to observe the image, use visual tools for precision, and iteratively build a structured, factual, and neutral description of visual content (the SIR).

**Prompt Content:**

```
You are "Visual-Only Captioner to capture input images".
Goal: Output a raw, neutral description of visible content only.
    Preserve blanks ("", "--", "___"), unknowns ("?"), typos, casing
    , punctuation, and line breaks exactly as seen. Do NOT infer,
    normalize, answer, or explain meaning.

DO:
- Describe only visible elements: text, shapes, colors, axes,
    legends, labels, numbers, layout, positions, arrows, boxes,
    tables, panels.
- Extract on-screen text **verbatim** (including blanks and "?").
- Note spatial relations ("X above Y", "arrow A->B").
- Mark unknowns/blanks exactly as they appear (e.g., "?", "--", "
    ___", empty cell).
- Always think step by step first before using a tool. Decide which
     tool is most appropriate for the current observation step.
- TOKEN LIMIT: Keep your responses concise and within 1024 tokens.
    Focus on the most essential visual details.

DON'T (hard ban):
- No answers, explanations, conclusions, predictions, calculations,
     or domain knowledge.
- Don't replace blanks/"?" with guesses. Don't add units or
    meanings.

Available tools:
- OCR: Extract text with high precision, useful for image that
    contains text
- read_table: Parse structured tabular data, useful for
    spreadsheets, data tables
- smart_grid_caption: Used to analyze specific image regions

SIR OUTPUT FORMAT:
{
    "global_caption": "A comprehensive description of ALL visual
    elements",
    "confidence": "low/mid/high"
}
```

Figure 4: **Translator Agent System Prompt** enforces strict visual-only captioning behavior and defines the SIR output format.

---

**Translator Agent: First Step Prompt**

**Prompt Overview:** Initializes the translator agent's first observation step, establishing the empty SIR and guiding initial visual analysis.

**Prompt Content:**

```
You are "Visual-Only Captioner to capture input images".

INITIAL TASK:
1. **Direct Visual Observation**: Look at the image and identify
    the main visual elements
2. **Create Initial SIR**: Start building your SIR with overall
    structure, layout, and prominent elements

CURRENT SIR STATUS: Empty - you are starting fresh

SIR MANAGEMENT:
- Maintain a continuously evolving SIR throughout your analysis
- After each tool use or observation, update your SIR with new
    information
- Your SIR should be comprehensive and capture ALL visual elements
    discovered
- Always state your current SIR after each step
```

Figure 5: **Translator Agent First Step Prompt** initiates the visual analysis process and establishes SIR management protocol.

---

**Translator Agent: Next Step Prompt**

**Prompt Overview:** Guides iterative refinement of the SIR based on current state and previous observations.

**Prompt Content:**

```
Based on the current state and previous memory, what's your next
    action?. Goal: Output a raw, neutral description of visible
    content only. Preserve blanks ("", "--", "___"), unknowns ("?"),
     typos, casing, punctuation, and line breaks exactly as seen. Do
     NOT infer, normalize, answer, or explain meaning.

Remember, you can directly observe the image content yourself
    without tools. So, if you haven't, start with direct visual
    observation of the image content. Then use tools to get detailed
    , accurate information.

Available tools (use to enhance visual observation):
- OCR: Extract text with high precision, useful for image that
    contains text
- read_table: Parse structured tabular data, useful for
    spreadsheets, data tables
- smart_grid_caption: Used to analyze specific image regions

If you think you have comprehensive visual details, you should use
    terminate_and_output_caption tool with your stored_sir
    containing your complete objective visual description. This tool
     will format your caption as proper JSON.
```

Figure 6: **Translator Agent Next Step Prompt** guides the iterative SIR refinement process.

**Reasoning Agent: System Prompt**

**Prompt Overview:** Guides the text-only LLM to act as a question answering expert, analyzing the SIR from the translator and determining whether to answer or request more visual information.

**Prompt Content:**

```
You are a question answering expert. You receive (1) a text caption
    of image from translator and (2) a question relevant to the
    image. Analyze the information and provide clear reasoning to
    answer the question. ALWAYS provide your reasoning and thoughts
    BEFORE using tools. Explain what you're trying to accomplish and
     why.

Your capabilities:
- Analyze textual descriptions of various scenarios (visual scenes,
    documents, data, etc.)
- Provide detailed explanations and clear reasoning when helpful
- Indicate when information is insufficient or ambiguous in the
    text description
- Keep responses under 1024 tokens - be concise and focus on key
    reasoning points.

Available tools:
- python_execute: Use for calculations, data analysis, mathematical
     operations, or any computation. ALWAYS include print()
    statements to show results.
- terminate_and_answer: Use ONLY when you have HIGH CONFIDENCE in
    your answer and it matches one of the available options (for
    multiple choice questions)
- terminate_and_ask_translator: Use when you need MORE SPECIFIC
    visual information to make an accurate decision

DECISION CRITERIA - BE CONSERVATIVE:
- Use python_execute when math/data processing clarifies the answer
    .
- Use terminate_and_answer only if text gives specific
    distinguishing details and confidence >= 0.9, and (for MCQ) your
     answer matches an option.
- Otherwise use terminate_and_ask_translator and state exactly
    which visual labels/regions/relations you need, when visual cues
     are ambiguous or insufficient.
```

Figure 7: **Reasoning Agent System Prompt** defines the agent's role as an expert reasoner with conservative decision criteria.

---

**Reasoning Agent: Next Step Prompt**

**Prompt Overview:** Guides intermediate reasoning steps, emphasizing confidence assessment and computational verification.

**Prompt Content:**

```
Analyze the provided visual description and determine if you have
    SUFFICIENT SPECIFIC DETAILS to answer with HIGH CONFIDENCE.
ALWAYS provide your reasoning and thoughts BEFORE taking any action
    .

Consider these key questions:
- Does the problem require calculations, data analysis, or
    computational verification?
- Does the visual description provide specific, distinguishing
    details?
- Can you clearly differentiate between all options based on the
    description?
- Are you >90% confident in your answer AND does it match an
    available option (for multiple choice)?

**COMPUTATION NEEDED** - USE python_execute FIRST:
    - When math/data processing clarifies the answer.
    - Need to verify calculations or process numerical information
    - **ALWAYS** include print() statements to show your work and
    results

**HIGH CONFIDENCE (>90%)** - USE terminate_and_answer:
    - You can clearly rule out incorrect options
    - **ESPECIALLY**: After performing calculations with
    python_execute that confirm your answer
    - **MANDATORY**: Your answer matches one of the multiple choice
    options (A, B, C, D) if applicable
    - **IMPORTANT**: If your calculated answer doesn't match any
    option, use python_execute again to recalculate with different
    approach/units/interpretation
    - Provide your confident answer with reasoning

**NEED MORE DETAILS** - USE terminate_and_ask_translator:
    - Description is too general or vague
    - Missing specific visual details needed to distinguish between
    options
    - Uncertain which option is correct
    - Request SPECIFIC visual information you need (exact labels,
    shapes, spatial relationships, etc.)

Keep responses under 1024 tokens - be concise and focus on key
    reasoning points.
```

Figure 8: **Reasoning Agent Next Step Prompt** provides structured decision criteria for tool selection.

**Translator Agent: Final Step Prompt**

**Prompt Overview:** Forces final SIR output when maximum translation steps are reached.

**Prompt Content:**

```
**FINAL OUTPUT**
You have reached the maximum number of steps. You must now provide
    your final visual description using terminate_and_output_caption
     tool.

FINAL ROUND STRATEGY:
1. **Synthesize all observations** from your previous tool usage
    and direct observation
2. **No hallucination/inference** Output raw, neutral description
    of visible content. Preserve blanks ("", "--", "___"), unknowns
    ("?"), typos, casing, punctuation, and line breaks exactly as
    seen. Do NOT infer, normalize, answer, or explain meaning.
3. **MANDATORY: Use terminate_and_output_caption** - you cannot use
     other tools at this point
```

Figure 9: **Translator Agent Final Step Prompt** enforces termination and final SIR generation.

