# OpenReview forum: "SeeingEye: Agentic Information Flow Unlocks Multimodal Reasoning in Text-Only LLMs"
_ICLR.cc/2026/Conference — ICLR 2026 Conference Withdrawn Submission_

### Official Review · Reviewer_dnrX · 2025-10-17

**Soundness:** 2
**Presentation:** 2
**Contribution:** 2
**Rating:** 2
**Confidence:** 3

**Summary:**

The paper introduces SeeingEye, a framework that enables text-only LLMs to improve their multimodal reasoning capabilities. It consists of two main agents: a Translator (VLM) and a Reasoner (LLM). The VLM-based Translator utilizes tools (e.g., OCR, cropping) to generate structured data called SIR from the image in a VQA question. The LLM-based Reasoner then extracts information from the provided textual data to answer the question. If the SIR does not contain sufficient information to solve the problem, the process continues iteratively, guided by feedback from the Reasoner. Experimental results demonstrate that the framework enhances performance by integrating visual and textual reasoning in two distinct yet interactive stages, coordinated through Agentic Information Flow.

**Strengths:**

•	The organization of the paper is good, contains examples, and explanations.

•	The structure of the framework is clearly explained.

**Weaknesses:**

•	The authors aim to unlock multimodal reasoning in Text-only LLMs; however, the LLM-based reasoner is just extracting textual information from the structured text. The main process is done with a VLM-based translator, which determines the visual part of the VQA question and converts it to a textual result. So, the success of the process depends on the translator-based VLM, which makes it not purely LLM reasoning.

•	The authors state that “the results highlight a scalable pathway to advanced multimodal reasoning”, however, they tested the SeeingEye framework only on knowledge-intensive VQA benchmarks. It would be beneficial to clarify which specific types of reasoning (e.g., relational, spatial, analogical) the study is designed to target, in order to better contextualize the scope of its capabilities.

•	The experiment can be more complete. The current comparison with other frameworks is limited, and the range of datasets used is relatively narrow. Apart from the MMMU variations, the results appear comparable to those of existing VLMs or the OpenManus framework. Including a broader set of benchmarks would help assess the robustness and generalizability of the proposed approach.

**Questions:**

• The paper refers to a “Translation First” framework in some sections. Should this term refer to SeeingEye? If not, could the authors clarify what is meant by “Translation First” and how it relates to the proposed system?

• In several parts of the paper, SeeingEye is described as a "model." Could the authors clarify whether it is a model or a framework?

• What specific tools are implemented within the framework (e.g., OCR, cropping)? For each, what type of problem is it intended to solve?

• The authors mention that Qwen2.5-VL is used as a visual analysis tool. Could the authors clarify which tasks this model is used for within the framework?

• In the Translator Agent component, how is it determined whether the structured image representation (SIR) provides sufficient information? How does this decision process differ from the evaluation and feedback mechanism in the Reasoning Agent? (Fig 1)

• Can the authors provide an error analysis? Specifically, what is the source of incorrect answers—does the issue lie in the Translator Agent or the Reasoning Agent?

• The OCR-BenchV2 results for SeeingEye appear similar to the baseline Qwen2.5-VL model. What factors might explain why the framework does not show improved performance in this case?

• The reported MMMU-val score of GPT-4o-mini in the paper differs from the official results provided by OpenAI and the MMMU benchmark website. Could the authors explain this discrepancy?
https://openai.com/index/gpt-4o-mini-advancing-cost-efficient-intelligence/
https://mmmu-benchmark.github.io/#leaderboard

• Section 5.2 aims to demonstrate the effectiveness of the Agentic Information Flow. However, there are differences between the frameworks being compared—for instance, SeeingEye uses structured representations (SIR), whereas OpenManus does not. How do the authors ensure that the performance difference is due to the information flow mechanism and not the use of SIR?

• How does SeeingEye perform beyond the third iteration of its reasoning loop? Does accuracy decline with more iterations? Have the authors identified an optimal number of iterations?

• Across the datasets used, on which types of tasks does the SeeingEye framework perform best? Are there task types where the system underperforms or may require further improvement?

• How is the proposed framework cost-effective in terms of compute, model size, or inference time?

• Figure 1 presents additional multi-agent baselines such as GeRea and MM-Reasoner. However, these are not included in the experimental comparison.

Suggestions:

• The performance impact of the Agentic Information Flow is described in both the Discussion and Main Results sections, which duplicates the explanation and may lead to redundancy.

---

### Official Review · Reviewer_Rzzg · 2025-10-28

**Soundness:** 2
**Presentation:** 3
**Contribution:** 2
**Rating:** 2
**Confidence:** 4

**Summary:**

In this paper, the authors propose SeeingEye, a modular framework that unlocks multimodal reasoning in text-only LLMs through an agent-based small VLM translator. Namely, they observe that LLMs (reasoner) are good at reasoning, and VLMs (translator) are good at perception. Therefore, they combine the strengths of them to improve visual reasoning. SIRs are used to convey infomation.

**Strengths:**

1. This paper aims to enhance visual reasoning capabilities, which is an important and widely recognized problem in the research community.

2. The paper is overall easy to follow.

3. The references are relatively comprehensive.

**Weaknesses:**

Although I believe this paper is technically solid overall, I have several concerns:

1. On novelty. The paper focuses on improving visual reasoning by **constructing prompting scaffolds** to enhance reasoning ability. However, similar ideas have been extensively explored in both the LLM and VLM literature, including but not limited to Prism [1]. While earlier designs may have been less sophisticated, the underlying idea is largely similar. A common issue with this line of work is that as VLMs themselves become more capable, the effect of the scaffolding diminishes. For instance, if the vision translator were replaced with o3, would the proposed framework still remain so effective?

2. On evaluation. I have concerns about the evaluations presented in Table 1. Since this work is not the first to introduce an agentic pipeline for visual reasoning, the performance of the four end-to-end VLM models serves only as a reference point (though their inclusion is appreciated). The focus should instead be on comparing against other agentic baselines. V* is a reasonable baseline, but it is relatively early work whose underlying VLM is significantly weaker than current models, thus limiting the interpretability of the numerical comparison. As for OpenManus, since the proposed pipeline appears to be built upon it, a detailed comparison is essential. However, the paper lacks a clear description of how the different tools are integrated into the OpenManus baseline. Additionally, it is unclear whether the baseline and the proposed method consume comparable computational resources. **If the proposed pipeline involves more reflective or iterative steps, while OpenManus performs only a single-pass inference, the comparison would not be fair.**

3. On SIRs. Although the authors claim that SIRs are critical to the approach, there is no experiment demonstrating how SIRs differ from simpler formats such as plain text or JSON for information exchange.

4. On reproducibility and credit. Although the supplementary materials provide code, there are no clear instructions on how to run it. Moreover, the code reveals that the work is largely based on OpenManus, yet the main text does not sufficiently credit this, giving the impression that OpenManus is merely treated as a regular baseline.

[1] Yuxuan Qiao, Haodong Duan, Xinyu Fang, Junming Yang, Lin Chen, Songyang Zhang, Jiaqi Wang, Dahua Lin, & Kai Chen. (2024). Prism: A Framework for Decoupling and Assessing the Capabilities of VLMs.

**Questions:**

1. I hope the authors can address the concerns mentioned in the four weaknesses above.

2. It would be helpful to fix the formatting in the appendix, as some pages (e.g., page 12) are not well presented.

3. In Section 5.1, it would be beneficial to include results using a larger version of Qwen3 to further support the scaling argument.

4. Section 5.1 mainly compares SeeingEye with different LLMs. Would a similar scaling effect be observed if SeeingEye with different VLM sizes were used?

---

### Official Review · Reviewer_qjXh · 2025-10-31

**Soundness:** 2
**Presentation:** 2
**Contribution:** 2
**Rating:** 2
**Confidence:** 3

**Summary:**

This paper proposes to unlock multimodal reasoning in a text-only LLM by creating a cooperative framework with a visual VLM. Now in contrast to having a single LVLM context which expands as the ‘dialog’ goes on, the paper proposes to have a structural intermediate representation which is passed and updated between the two reasoning and the visual agent. Both agents are created by prompt engineering for the task. The paper also proposes an algorithm for the control flow of the dialog itself.

Results demonstrate that their method, which is based on Qwen 2.5-VL-3b outperforms the larger Qwen 2.5-VL 72b parameter model and also outperforms GPT-4o-mini on MMMU and MMMU-Pro, but has less convincing results on OCR-Bench and MIA-Bench (Table 1).

**Strengths:**

* Separating sub-tasks makes sense but is also not novel. Previous published works in this direction are [Socratic Models, Zeng et al., ICLR'23] and [HAMMR, Castrejon et al., NeurIPS workshop 2024] and if you just go to LLMs using visual tools there are the seminal [VisProg CVPR’23] and [ViperGPT ICCV’23] papers which had quite a few follow-up papers.
* Results seem fine w.r.t. the used baselines on MMMU and MMMU-Pro

**Weaknesses:**

* Giving LLMs visual capabilities with tool calls is not new. See references in point 1 of the strengths.
* As baselines, only vanilla Qwen and GPT-4o are used. I do not see any attempts to use contemporary techniques with these models such as Chain-of-Thought versions or LLMs with tool calls. Just the vanilla versions.
* The other baseline is OpenManus. There are currently many agentic frameworks around and it is hard to estimate how good a baseline this is by just reading this paper.
* In their experiments the authors claim that the difference between OpenManus and their method is primarily caused by the structured representation which is being passed between agents. The main problem here is that the prompts in this paper were heavily optimized to handle this structured representation, while it remains unclear how much effort was put into the OpenManus baseline. Without understanding how good OpenManus is, it is hard to estimate the value of this result.
* There are several recent benchmarks specifically designed for LLMs with visual inputs: V*Bench, VisualProbe, HR-Bench, and MME-RealWorld come to mind. No results or comparisons are done on these benchmarks.
* Results on OCR-Bench are unconvincing: On OCR-BenchV2 I see an improvement from 33.33 for Qewn-2.5-VL-3b to 33.99 for the proposed method (using this model). This improvement seems not significant. On MIA-Bench, Qwen-72b is much much better while OpenManus is close (82.4 for OpenManus and 84.1 for this paper). But again I would expect some comparison with more agentic/tool-use based methods.

**Questions:**

The problem which I have with this paper is twofold:
* Separating reasoning from the vision component is not really novel and has been done before.
* I am not convinced that this paper shows state-of-the-art results:
** I do not find the baselines convincing given the activity in the LLMs+tools space.
** It remains unclear how the OpenManus results should be valued w.r.t. the state-of-the-art.
** Only MMMU results show some improvements. Results on OCR-Bench and MIA-Bench are unconvincing.
** Several more recent benchmarks specifically introduced for more difficult multimodal reasoning tasks are missing.

---

### Official Review · Reviewer_Vpvh · 2025-11-04

**Soundness:** 3
**Presentation:** 2
**Contribution:** 2
**Rating:** 4
**Confidence:** 3

**Summary:**

This paper presents SeeingEye, a framework that combines multimodal agent and a stonger text-only agents to achieve multimodal reasoning. SeeingEye utilizes a vision-language model (Qwen2.5-VL) acting as a vision-language translator, and a text-only LLM (Qwen3) acting as the reasoning agent. Besides, this paper proposes SIR, standing for structured intermediate representation, so that both agents can communication with each other with a pre-defined format. Experimental results show that SeeingEye outperforms both Qwen2.5-VL models and two modular frameworks on several visual question answering benchmarks.

**Strengths:**

- The idea of building multi-agent systems for multimodal reasoning is a compelling direction. It is a natural path toward a form of collective intelligence where specialized agents collaborate to solve complex problems. This paper effectively explores how to combine a perception-focused agent (the VLM Translator) with a cognition-focused agent (the text-only LLM Reasoner), which is a promising paradigm for building more capable and interpretable AI systems.
- The experimental results convincingly support the paper's central claim. The fact that a combination of smaller, specialized models (3B VLM + 8B LLM) outperforms a much larger, monolithic 32B VLM on reasoning-heavy benchmarks is a significant finding. It provides strong evidence that a well-designed modular architecture can be more parameter-efficient and effective than simply scaling up end-to-end models.

**Weaknesses:**

- Motivation: The paper's premise is that text-only LLMs possess reasoning capabilities superior to those found in monolithic VLMs, thus motivating the need for decoupling. However, this motivation is becoming less convincing, as state-of-the-art VLMs (e.g., Qwen3-VL) are typically built upon the most capable LLMs available at the time of their creation. The argument implicitly assumes a gap in reasoning ability that may not exist, or at least is not sufficiently justified.
- Prompting LLMs to do multi-agent communication is an established concept, so the core novelty of this work might be on the Structured Intermediate Representation (SIR) that defines how they communicate. However, SIR is not well-defined. The SIR is presented as the central channel for communication, but its schema is not formally defined.
- The experiments, while showing strong top-line numbers, lack depth in analysis. The benefit of multi-round interaction (Table 4) is an expected outcome for any iterative multi-agent system. The discussion would be significantly strengthened by providing scaling behaviors across diverse tasks and more iterations.
- Other presentation issues. For instance, the paper sometimes writes "3B" and sometimes "3b"; Line-285: "our SeeingEyeframework. Our" -> "our SeeingEye framework. Our"; Inconsistent style between Table 2 and Table 3; In table 3, "SeeingEye(Ours)" -> "SeeingEye (Ours)"

**Questions:**

Please refer to the weakness section.

---

### Note · Authors · 2025-12-03

I have read and agree with the venue's withdrawal policy on behalf of myself and my co-authors.